# Unified Lexical Representation for Interpretable Visual-Language Alignment

**Yifan Li**[1][*]   **Yikai Wang**[1][†]   **Yanwei Fu**[1][‡]   **Dongyu Ru**[2]
**Zheng Zhang**[2]   **Tong He**[2][†]
[1]Fudan University   [2] Amazon Web Services
yifanli23@m.fudan.edu.cn,  yi-kai.wang@outlook.com,
yanweifu@fudan.edu.cn,  {rudongyu, zhaz, htong}@amazon.com

## Abstract

Visual-Language Alignment (VLA) has gained a lot of attention since CLIP's groundbreaking work. Although CLIP performs well, the typical direct latent feature alignment lacks clarity in its representation and similarity scores. On the other hand, lexical representation, a vector whose element represents the similarity between the sample and a word from the vocabulary, is a natural sparse representation and interpretable, providing exact matches for individual words. However, lexical representations are difficult to learn due to no ground-truth supervision and false-discovery issues, and thus requires complex design to train effectively. In this paper, we introduce LexVLA, a more interpretable VLA framework by learning a unified lexical representation for both modalities without complex design. We use DINOv2 as our visual model for its local-inclined features and Llama 2, a generative language model, to leverage its in-context lexical prediction ability. To avoid the false discovery, we propose an overuse penalty to refrain the lexical representation from falsely frequently activating meaningless words. We demonstrate that these two pre-trained uni-modal models can be well-aligned by fine-tuning on the modest multi-modal dataset and avoid intricate training configurations. On cross-modal retrieval benchmarks, LexVLA, trained on the CC-12M multi-modal dataset, outperforms baselines fine-tuned on larger datasets (e.g., YFCC15M) and those trained from scratch on even bigger datasets (e.g., 1.1B data, including CC-12M). We conduct extensive experiments to analyze LexVLA. Codes are available at https://github.com/Clementine24/LexVLA.

## 1   Introduction

The development of Vision-Language Alignment (VLA) models has made great progress [43, 48, 22, 26, 6] since the innovative CLIP [34] effectively learns a shared latent space where text and image are well-aligned. This progress has also boosted related fields like vision-language foundation models [20], multi-modal understanding [18], and text-conditional generation [35]. However, CLIP's latent features pose challenges of interpretability issue for analyzing individual factors' impact. Additionally, CLIP's visual model struggles to learn patch-level features, and its text model are trained based on incomplete and biased captions. These challenges reduces its overall effectiveness.

---

[*]Work completed during internship at AWS Shanghai AI Lab.

[†]Corresponding authors.

[‡]Dr. Fu is with School of Data Science in Fudan, Fudan ISTBI-ZJNU Algorithm Center for Brain-inspired Intelligence, Shanghai Key Lab of Intelligent Information Processing, and Technology Innovation Center of Calligraphy and Painting Digital Generation. Ministry of Culture and Tourism. China.

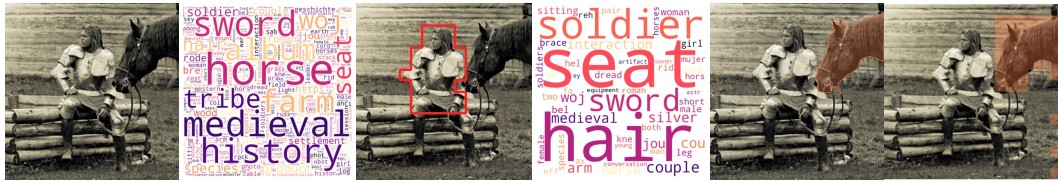

Figure 1: LexVLA can generate a lexical representation of the input image (the first word cloud figure), or to pick some patches of the image for local lexical information representation (the second word cloud figure, with the seleted patches boxed in red), and to select the most relevant patches of the image given the text content (the rightmost figure, with caption 'horse', with the second-to-last figure is the ground-truth mask).

On the other hand, the lexical representation is known for its clarity as each dimension corresponds to the similarity between the input and a specific word/token from the vocabulary. Additionally, lexical representation can naturally be made sparse, which means it's efficient for tasks like search or indexing in large-scale retrieval tasks. However, learning lexical representation is difficult. The embedding vector is much larger than the CLIP latent vector as the vocabulary size is usually much larger than CLIP's feature dimension. This poses several challenges: the lack of precise supervision signals; the incomplete and biased text supervision, and the high-dimensional embedding space with low-dimensional targets. Thus using lexical representations in CLIP-style VLA models is tough.

Previous lexical representation learning approaches tries to utilize strategies including $\ell_1$ regularization [46, 3] to direct encourage sparisity or use ReLU activation [27] to remove negative scores [12, 5, 25] or directly regularize the floating-point operations [31]. However, these intricate training configurations usually introduce extra training difficulty and additional regularization such as log saturation [12, 25] and bag-of-words penalization [48] which potentially limits the VLA capacity.

In this paper, we propose LexVLA, a simple yet comprehensive framework for learning a unified lexical representation in the CLIP-style contrastive training pipeline, facilitating the VLA. LexVLA enjoys the following merits in addressing the challenges above. **1)** We use single-modal pre-trained models for their unique benefits, including DINOv2 [30] as vision model for its local-inclined features, and Llama 2 [40] as text model to learn lexical representations through in-context prediction tasks instead of traditional caption embeddings. **2)** We suggest a unified *vocabulary* with distinctive *codebooks* for each modality to maintain the strengths of single-modal pre-trained models by refrain them from learning the same feature space. This allows us to create better VLA models with less multi-modal training data and simpler setups. We also introduce an overuse penalty to prevent excessive activation of irrelevant tokens, improving upon the FLOPs loss [31] which only promotes sparsity. **3)** We incrementally train LexVLA to stay aligned with pre-trained models. For text, we fine-tune Llama 2 with LoRA adapter[14] and keep its codebook frozen. For vision, we freeze DINOv2 and train a projector with a self-attention layer and two multi-layer perceptions to the vision codebook, initialized using the text codebook. **4)** We introduce PatchDis, a metric for patch-level alignment in visual features, evaluating patch-level classification tasks with image patch features and category text tokens learned by VLA models. We propose PatchDis as a patch-level interpretability metric for VLA models not trained on fine-grained tasks like segmentation or detection.

Formally, our LexVLA employs a unified lexical vocabulary with distinct codebooks for text and image modalities, utilizing bi-encoders to individually encode inputs into lexical representations. These encoders, initially set with single-modal pre-trained models, undergo incremental fine-tuning via the standard contrastive learning objective, with the integration of an overuse penalty to promote sparsity and mitigate token over-activation. Emprically, we show the effectiveness of LexVLA on two zero-shot cross-modal retrieval benchmarks. Trained on the CC-12M dataset, it outperforms baselines fine-tuned on larger datasets like YF100m-CLIP + CC-12M and those trained from scratch on even larger datasets totaling 1.1B data, including CC-12M. We provide detailed analysis of LexVLA.

Our contributions can be listed as follows:

- We emphasize utilizing single-modal pre-trained models for vision-language alignment tasks to benefit from their unique properties that cannot learned by contrastive objectives. To exemplify, we use DINOv2 for its local-inclined features and Llama 2 for its in-context capacity.

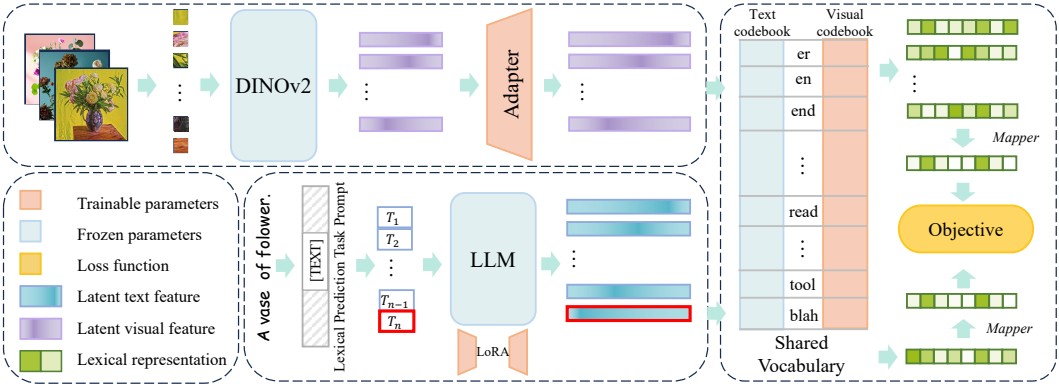

Figure 2: Framework of LexVLA. We learn a unified lexical representation with distinct codebooks for text and visual modalities. For the image, we adopt the frozen DINOv2, learn an adapter and a mapper to get visual lexical representation. For the text, we use LoRA to fine-tune the Llama 2 on in-context lexical prediction task, followed with a mapper to get the text lexical representation. We initialize codebooks as Llama 2's codebook, freeze the text codebook while fine-tuning the visual codebook. We train LexVLA with the standard contrastive objectives along with the proposed overuse penalty to encourage sparsity while preventing meaningless activation.

- We effectively learn a unified lexical representation with unique codebooks for vision and language modalities to refrain from the weakening of pre-trained capabilities.
- We propose an overuse penalty to encourage sparse embedding and prevent meaningless activation.
- We enjoy superior retrieval performance with less multi-modal training data, and achieve better patch-level interpretable VLA model with global supervision signals, quantitatively surpassing CLIP-style and lexical methods with our proposed patch-level interpretability metric PatchDis.

## 2 Related works

**Vision-Language Alignment**    VLA has been a challenging problem in deep learning. The innovation of CLIP [34] has inspired a plethora of research efforts in combining visual and language modalities [21, 44, 26, 37, 6, 43]. With the rise of large language models (LLMs) [1, 40, 38], it's become common to use CLIP to get visual data from images and combine it with pretrained LLMs[24, 8, 20, 2, 49]. However, recent studies have identified visual encoders as potential bottlenecks in these models, posing issues like identifying actions and spatial relationships [41, 15], ignoring attributes and states [45, 10], introducing hallucination problems [39], etc. Nevertheless, progress in using other visual models instead of CLIP has been slow. Tong et al.[39] discovered a drop in instruction-following ability when integrating DINOv2[30] into LLaVA's [24] structure. To address this, they combined features from CLIP and DINOv2. Similarly, Zhai et al. [47] used a locked DINO for contrastive tuning, which necessitated significant aligned data and computational resources. In contrast, we use a locked DINOv2 for lexical VLA, demonstrating the superiority over CLIP with smaller amount of multi-modal training data. It is important to note that our work differs significantly from recent zero-shot cross-modal retrieval models, such as BEiT-3 [42]. Our approach emphasizes developing specific lexical representations for images and text, whereas general retrieval models lack this interpretable representation.

**Multi-modal lexical representation**    Lexical representation is popular in information retrieval [12, 13] thanks to its interpretability and efficiency. However, adapting this approach to vision-language models is challenging because of the differences between the two modalities. Previous approaches rely on complex training stages (two [25] or three [5]), large scale multi-modal training data [5] to train effectively and additional Bag-of-Words (Bow) restrictions [25, 48] to prevent semantic misalignment. In contrast, we in this paper propose a single fine-tuning stage to align uni-modal pre-trained models in the output lexical space with less multi-modal training data and without BoW restriction. In this paper, our model shall be equipped with the generalizable ability for zero-shot cross-modal evaluation, rather than fine-tuning on each specific target cross-modal training dataset.

Additionally, it is important to note that lexical representation differs fundamentally from codebook strategies, as highlighted in [11]. While our aim is to develop interpretable lexical representations for images and visual patches at the vocabulary level, offering greater flexibility, codebook strategies focus on creating a unified yet non-explicit semantic code for images and text at the codebook level, which lacks the desirable traits of interpretability and high retrieval efficiency.

# 3  LexVLA

**Problem setup**   Our target is to learn the lexical alignment between text and image modalities. Typically, the lexical representation $s_i \in \mathbb{R}^V, i \in \{\text{img, txt}\}$ is a score vector, whose element indicates the similarity between the sample and the corresponding word from the vocabulary $\mathbf{V}$ with $V = |\mathbf{V}|$. The contrastive lexical alignment aims to train the lexical encoders for each modality such that the similarity $\langle s_{img}, s_{txt} \rangle$ for positive image-text pairs are maximized and that for negative pairs are minimized, while preserving the correct lexical correspondence with the vocabulary.

**Overview**   Our model, as in Fig. 2, learns a single lexical representation for VLA (Sec. 3.1). We encode each modality separately using uni-modal encoders (Sec. 3.2) adapted in Sec. 3.3.

## 3.1  Lexical representation

**Vocabulary**   Inspired by tokenization techniques [36, 19], we initialize with the tokenizer vocabulary and refine it by removing unused and meaningless tokens, reducing the size from 32,000 to 17,149.

**Codebooks**   The one-hot embedding is semantic-less and assume the same distance between different words, which is obviously not a good embedding. In this paper, we use the output codebook of language models as a well pre-trained and semantic-rich codebook, inducing lexical codes of 4096-dim vectors. However, the text codebook is not optimized for visual features. Basically, during the pre-training of language model, the visual world is not observable, thus we may have a gap between text-only embeddings and the visual-embeddings. To this end, we propose to use the same *vocabulary* $V$ but a unique *codebook* $\mathbf{Z}_i, i \in \{\text{img, txt}\}$ for text and image modalities, respectively. As the language codebook is pre-trained on large corpus, we freeze $\mathbf{Z}_{txt}$ to enjoy the well-trained text embeddings. $\mathbf{Z}_{img}$ is initialized with the weights of $\mathbf{Z}_{txt}$ and fine-tuned along with the training of lexical encoders to enable the adaptation for visual features.

**Sparse representation**   Thanks to the inherent similarity measurement in lexical representations, it is natural to turn a dense output vector into a sparse lexical representation. Typically we could utilize the following strategies: 1) Top-k thresholding: select the most important $k$ items and discard the rest; 2) Value thresholding: keep items with value above the threshold and discard the rest. In this paper, we use the value thresholding strategy. We select items with values greater than $1/\sqrt{V}$, which means we consider only the items with above-average signals as informative.

## 3.2  Lexical encoder

Given the image as $\boldsymbol{x}_{img}$ and text as $\boldsymbol{x}_{txt}$, the lexical representation is achieved by:

$$\boldsymbol{s}_i := e_i(\boldsymbol{x}_i) = h_i \circ g_i \circ f_i(\boldsymbol{x}_i), i \in \{\text{img, txt}\}. \tag{1}$$

Without loss of generality, we ignore the subscript $i$ in the following definitions unless otherwise specified. We split the lexical encoder in three stages: 1) The single-modal pre-trained feature extractor $f$, takes as input $\boldsymbol{x}$ and as output a feature sequence $\boldsymbol{y} := f(\boldsymbol{x}) \in \mathbb{R}^{n \times d_i}$, where $n$ is the sequence length and $d_i$ is the modal-dependent feature dimension; 2) The projector $g$, projects $\boldsymbol{y}$ to the lexical feature sequence $\boldsymbol{z} := g(\boldsymbol{y}) \in \mathbb{R}^{n \times d}$, where $d$ is the lexical feature dimension; 3) The mapper $h$, maps $\boldsymbol{z}$ to the lexical representation $\boldsymbol{s} = h(\boldsymbol{z}) \in \mathbb{R}^V$.

In this paper, we restrict the lexical representation to be non-negative and $\ell_2$-normalized, i.e., $\|\boldsymbol{s}\|_2 = 1, s_i \geq 0$. Different modalities share the same $V$ with a unique codebook $\boldsymbol{Z}_i \in \mathbb{R}^{V \times d}, i \in \{\text{img, txt}\}$.

### 3.2.1 Lexical text encoder

**Captioning?** In VLA training data pair, the text is typically a caption of the corresponding image. However, it is well-known that caption only captures a partial observation of the semantics in the image, and cannot serve as a perfect target to represent the image. As it is the only accessible ground-truth signal, previous approaches mainly train the text encoders to represent the caption embedding via the output hidden state [9, 48, 12, 25] or [CLS] token embeddings [34], and use it as the target for the corresponding image. Hence, the learned alignment is biased and leads to false-elimination of image patterns that are not observed in the captions.

**Predicting!** Inspired by the powerful Large Language Models (LLMs), we would like to investigate *can we unleash the inherent knowledge of LLMs for lexical representation?* The answer is yes.

As the auto-regressive LLMs [1, 40] are not naturally to summarize the previous tokens, it is irrational to directly use the predicted token as the embedding for raw text inputs. A straightforward way to activate LLM is to use prompt like "summarize the sentence of [TEXT] in one word:" to activate LLM's summarization capacity and use the output token as the text embedding. But it is still the caption-style summarization, and cannot fully utilize the LLM.

**LLM as lexical predictor** Given the in-context learning capacity of LLM, we propose to adapt LLM as a lexical predictor. Specifically, our input for the LLM is:

> The focus of "The man is riding a white horse." lies on important words:"man", "riding", "white", "horse". The focus of "[TEXT]" lies on important words:

The [TEXT] is the input caption. Then we adopt the output token as the text embedding. This prompt includes two parts: 1) One in-context example to guide the LLM to identify the important words in the caption, adapting the LLM to lexical prediction task; 2) The question prompt to ask the LLM the important words of input caption. In this design, the output token of the LLM naturally performs the lexical prediction task to calculate the similarity between the input caption and every word in the vocabulary, and thus serves as a powerful proxy to get the lexical representation.

**Realization** we utilize Llama 2 [40] as our text encoder $f_{txt}$. The input text follows the above in-context prompts. We use the output of the predicted token embedding as $\boldsymbol{y}_{txt}$. As our lexical codebook is Llama 2's output codebook, we do not need $g_{txt}$, thus $\boldsymbol{z}_{txt} = \boldsymbol{y}_{txt}$. To implement $h_{txt}$, we first calculate dot-product attention between each text token $\boldsymbol{z}_{txt} \in \mathbb{R}^{1 \times d}$ and the text lexical codebook $\boldsymbol{Z}_{txt}$. Then we use the $elu1p$ activation [48] to transform the attention score into non-negative values and then normalize it, yielding global lexical representation $\boldsymbol{s}_{txt}$. Formally,

$$\boldsymbol{s}_{txt} = \text{Normalize} \circ elu1p(\boldsymbol{z}_{txt}\boldsymbol{Z}_{txt}^{\top}), \qquad (2)$$

where $elu1p(x) = (x + 1) \cdot 1_{\{x \geq 0\}} + e^x \cdot 1_{\{x < 0\}}$ and Normalize is the $\ell_2$ normalization operator. For word-level lexical representation, we directly use the word code in $\boldsymbol{Z}_{txt}$ followed by Eq. (2).

### 3.2.2 Lexical visual encoder

We adopt DINOv2 [30] as our visual backbone to implement $f_{img}$. Given input $\boldsymbol{x}_{img}$, we first flatten it into patches $\boldsymbol{x}_{img} = [\boldsymbol{x}_1, \ldots, \boldsymbol{x}_{n-1}]$ and input to $f_{img}$ to get

$$\boldsymbol{y}_{img} = f_{img}([\texttt{CLS}; \boldsymbol{x_{img}}]) \in \mathbb{R}^{n \times d}, \qquad (3)$$

where CLS is the "class" token in DINOv2. Subsequently, we implement $g_{img}$ with an adapter includes a self-attention layer and 2 multi-layer perceptions to post-process the DINOv2 features. To implement $h_{img}$, similar to the text encoder but in a more fine-grained manner, we first calculate dot-product attention between each image patch token $\boldsymbol{z}_{img,i} \in \mathbb{R}^{1 \times d}$ and the image lexical codebook $\boldsymbol{Z}_{img}$, followed by the $elu1p$ activation. In the end, we use max-pooling to aggregate patch representations and then normalize, yielding global lexical representation $\boldsymbol{s}_{img}$. Formally,

$$\boldsymbol{s}_{img} = \text{Normalize} \circ \text{Max-Pool} \circ elu1p(\boldsymbol{z}_{img}\boldsymbol{Z}_{img}^{\top}). \qquad (4)$$

The image patch lexical representation follows the same process but replace the max-pooling operator with selecting the corresponding patch location.

### 3.3 Train LexVLA

**Contrastive objective**  We use the InfoNCE loss [29] with learnable temperature $\tau$ for cross-modal retrieval over a batch of $N$ text-image pairs $(x_{img}, x_{txt})$ as the major objective:

$$\ell_{a2b} = -\frac{1}{N} \sum_{i=1}^{N} \log \frac{\exp(a_i b_i^\top / \tau)}{\sum_{j=1}^{N} \exp(a_i b_j^\top / \tau)}, (a, b) \in \{(x_{img}, x_{txt}), (x_{txt}, x_{img})\}, \tag{5}$$

**Overuse penalty**  In lexical representation learning, the FLOPs loss [31] is widely used to encourage the output sparsity. Given the modality-specified lexical representation matrix $\boldsymbol{S} = \{s_{i,j}\}, i \in \{1, \ldots, N\}, j \in \{1, \ldots, V\}$ in current mini-batch, the FLOPs loss is defined as:

$$\ell_{\text{FLOPs}} = \sum_{j=1}^{V} \bar{s}_{\cdot,j}^2 = \sum_{j=1}^{V} \left( \frac{1}{N} \sum_{i=1}^{N} s_{ij} \right)^2. \tag{6}$$

The FLOPs loss aims to reduce FLOPs to induce sparsity. But we noticed that sometimes this penalty pushes the encoder to take shortcuts that falsely activate rarely used and image-unrelated tokens. This semantic deviation leads to false activation and affects the interpretability of lexical representation.

To prevent using meaningless tokens too much, we adapt the FLOPs loss with a weighting method. Essentially, we aim to penalize tokens that are used too often across examples. We measure this via the normalized average activation value across the vocabulary, and propose the overuse penalty as:

$$\ell_{\text{overuse}} = V \sum_{j=1}^{V} \frac{\bar{s}_{\cdot,j}}{\sum_{k=1}^{V} \bar{s}_{\cdot,k}} \bar{s}_{\cdot,j}^2 = NV \sum_{j=1}^{V} \left( \sum_{i=1}^{N} s_{i,j}/N \right)^3 \Big/ \sum_{j=1}^{V} \sum_{i=1}^{N} s_{i,j}. \tag{7}$$

**Objective**  The full objective contains Eq. (5) and Eq. (7) for each modality:

$$\mathcal{L} = \ell_{t2i} + \ell_{i2t} + \lambda_I \ell_{overuse}^I + \lambda_T \ell_{overuse}^T \tag{8}$$

where $\lambda_I$ and $\lambda_T$ are two regularization weights for image and text representations, respectively.

**Incremental fine-tuning**  For text part, we adopt LoRA adapters [14] to fine-tune the Llama 2. For vision part, we freeze the DINOv2, and only train the projector and vision codebook.

### 3.4 PatchDis: interpretability metric

The core target of lexical representation is to achieve an interpretable feature representation for input modality. For the text input, the interpretability is straightforward to check. However, existing literature do not have a quantitative metric to measure the patch-level interpretability of visual lexical representation. In this paper, we propose the PatchDis metric to evaluate the patch-level discrimination tasks, inspired by the patch-level segmentation task but designed for models which are not trained from fine-grained alignment tasks like segmentation or detection.

Basically, PatchDis evaluates the classification task in the patch-level features. For any given VLA model, we use its text encoder to input the class names and obtain the text embeddings of all classes. Similarly, we input the testing image to the visual encoder to obtain all patch features. Then we could use the model-defined metric to calculate the similarity between each patch feature and the class embeddings, and predict the class of each patch from the largest similarity. Then for each class, we could aggregate all patches of one class as the patch-level prediction for the class. We use the ground-truth segmentation of the class as the target, and use mIoU as the metric to evaluate the patch-level interpretability of the VLA models in the zero-shot patch-level discrimination task.

## 4 Experiments

**Datasets**  We use CC-12M [4] for training, a dataset consisting of 12.4 million image-text pairs. We successfully download 9.2M pairs and use this subset as our training set. For evaluation, we use Flickr30k [33] and MSCOCO [23] to evaluate zero-shot cross-modal retrieval tasks.

Table 1: Zero-shot cross-modal retrieval. **Q** indicates variants of our LexVLA. CLIP[1] is the original CLIP [34]; results denoted by $(\cdot)^2$ are reported in VDR [48]; results denoted by $(\cdot)^3$ are reported in STAIR [5]. "Data" is the multi-modal alignment training data size; "Latent" means direct latent feature alignment methods; "Lexical" indicates lexical feature alignment methods. R@K, the recall ratio within top-K items.

| Setting | Model | Data | MSCOCO | | | | | | Flickr30k | | | | | |
| | | | image-to-text | | | text-to-image | | | image-to-text | | | text-to-image | | |
| | | | R@1 | R@5 | R@10 | R@1 | R@5 | R@10 | R@1 | R@5 | R@10 | R@1 | R@5 | R@10 |
| Latent | CLIP[2] | 15M | 20.8 | 43.9 | 55.7 | 13.0 | 31.7 | 42.7 | 34.9 | 63.9 | 75.9 | 23.4 | 47.2 | 58.9 |
| | FILIP[2] | 15M | 21.6 | 46.7 | 59.0 | 13.7 | 31.7 | 41.6 | 46.3 | 74.4 | 83.2 | 30.7 | 58.2 | 68.6 |
| | CLIP-BERT[2] | 15M | 23.9 | 47.8 | 60.3 | 13.6 | 33.8 | 45.1 | 44.1 | 71.2 | 80.7 | 27.8 | 54.7 | 65.9 |
| | DeCLIP[2] | 15M | 25.3 | 51.2 | 63.4 | 16.6 | 35.2 | 45.4 | 51.3 | 80.7 | 88.5 | 35.5 | 63.0 | 73.0 |
| | SLIP[2] | 15M | 27.7 | 52.6 | 63.9 | 18.2 | 39.2 | 51.0 | 47.8 | 76.5 | 85.9 | 32.3 | 58.7 | 68.8 |
| | ProtoCLIP[2] | 15M | 30.2 | 55.1 | 66.5 | 16.9 | 37.9 | 49.4 | - | - | - | - | - | - |
| | CLIP[1] | 0.4B | 52.4 | 76.7 | 84.6 | 33.1 | 58.4 | 69.0 | 81.8 | 96.2 | 98.8 | 62.1 | 85.6 | 91.8 |
| | CLIP[3] | 1.1B | 53.4 | 78.3 | 85.6 | 36.2 | 62.2 | 72.2 | 79.6 | 95.5 | 98.1 | 63.0 | 86.7 | 92.5 |
| Lexical | VDR[2] | 15M | 30.9 | 54.5 | 65.4 | 17.4 | 38.1 | 49.7 | 51.0 | 79.3 | 86.7 | 32.4 | 60.1 | 70.7 |
| | STAIR[3] | 1.1B | **57.7** | 80.5 | 87.3 | **41.4** | 65.4 | 75.0 | 81.2 | 96.1 | 98.4 | 66.6 | 88.7 | 93.5 |
| Lexical | **Q**(BoW) | 12M | 17.9 | 34.9 | 45.2 | 10.4 | 24.3 | 33.1 | 30.6 | 56.2 | 66.3 | 17.7 | 36.4 | 44.9 |
| | **Q**(CLIP) | 12M | 51.8 | 75.5 | 84.1 | 36.8 | 62.5 | 72.7 | 82.9 | 96.2 | 98.7 | 65.2 | 88.3 | 93.2 |
| | **Q**(FLOPs) | 12M | 56.2 | 80.0 | 87.4 | 39.0 | 65.7 | 75.6 | 84.2 | 96.6 | 98.7 | 67.4 | 89.4 | 94.1 |
| | **Q**(512) | 12M | 56.4 | 79.9 | 87.5 | 38.1 | 64.6 | 74.9 | **84.5** | 97.3 | 99.0 | 65.7 | 89.3 | 93.8 |
| | LexVLA | 12M | 55.4 | **80.6** | **88.3** | 39.8 | **66.3** | **76.2** | 83.9 | **97.5** | **99.1** | **67.8** | **90.2** | **94.2** |

**Implementation**    We use DINOv2 [30] base model and Llama 2 [40] 7B model as our backbones. We use Adam optimizer [17] with learning rate $5e-4$ and cosine decayed, batch size of 6,144, precision of BFloat16 for 12 epochs. We initialize $\tau$ as 0.07, and clip the logits larger than 100 as in [34]. We use 8 A100 GPUs of 40GB memory to train LexVLA. We quadratically warmup $\lambda$ in the fisrt 2k steps and then freeze as [31]. We set $\lambda_I$ as $5e-4$ and $\lambda_T$ as $1e-3$. The trainable parameters in LexVLA is 109 M in total, including 70M for vision codebook, 17M for vision projector (19.76% compared with DINOv2), 21M for Llama adaptor (0.30% compared with Llama 2).

## 4.1   Zero-shot cross-modal retrieval

**Evaluation**    We conduct experiments on zero-shot cross-modal retrieval tasks on Flickr30k and MSCOCO based on the splits in [16] following previous approaches. We use the R@K, recall ratio within top-K items, as the evaluation metric.

**Competitors**    We compare LexVLA with the latent feature alignment methods, including the original CLIP [34] model trained on different dataset scales and followup CLIP-style alignment models, including FILIP [43], CLIP-BERT [48], DeCLIP [22], SLIP [26], and ProtoCLIP [6]. We also compare with previous lexical representation learning models that focus on zero-shot cross-modal retrieval tasks, including VDR [48] and STAIR [5]. Note that all the competitors are trained by larger datasets. Specifically, the models trained on 15M data uses YFCC15M [21], and models trained on 1.1B data [5] contains several datasets including CC-12M.

**Variants**    For LexVLA, we report the following variants in addition to our final model: 1) **Q**(BoW), which selects only the nouns, adjectives, and non-auxiliary verbs in a caption (Bag-of-Words) instead of the LLM-based lexical predictor. 2) **Q**(CLIP), which uses the original CLIP visual model, instead of the proposed DINOv2 based encoder; 3) **Q**(FLOPs), which uses FLOPs loss as sparsity penalty. 3) **Q**(d), which uses the top-k thresholding strategy. We test $d = 512$, the same activated dimension as CLIP to make a fair comparison. The average activated dimension of the final LexVLA is 1081. Note that LexVLA surpasses CLIP with less activated values, see Fig. 5 for detailed analysis.

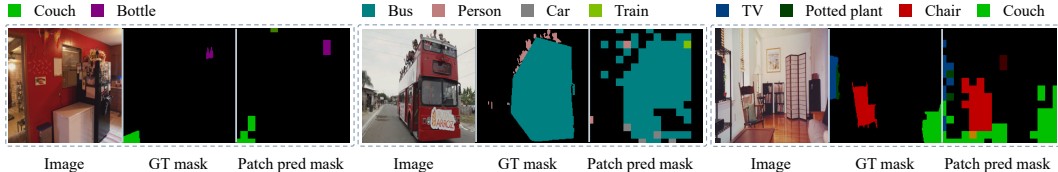

| Couch | Bottle | | Bus | Person | Car | Train | | TV | Potted plant | Chair | Couch |

Image     GT mask     Patch pred mask      Image     GT mask     Patch pred mask      Image     GT mask     Patch pred mask

Figure 3: PatchDis visualization. The same color indicates the same category. LexVLA correctly predicts the corresponding region, even for the small-scale objects, like the bottle in the first image.

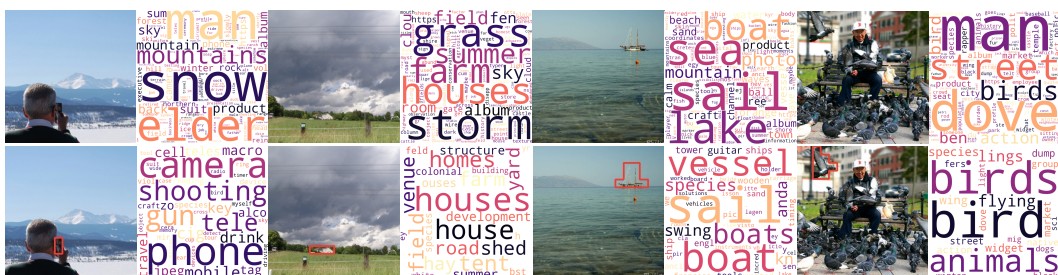

Figure 4: Visualization of the image lexical representation obtained by LexVLA. Larger word indicates larger lexical value. The first row represents the complete image, and the second row represent local patches (boxed in red). LexVLA learns a well-aligned lexical representation for both image and patches without local supervision.

**Results** We report the results in Table 1. **1)** LexVLA performs significantly better than both latent feature alignment methods and previous lexical alignment methods when trained on a similar amount of multi-modal examples (12M vs. 15M). This shows the effectiveness of LexVLA. **2)** Compared to CLIP trained on much larger multi-modal datasets (0.4B and 1.1B), LexVLA still performs better, even when using the same activated feature dimension as CLIP. This further demonstrates LexVLA's efficiency in text-image alignment with much less training data. **3)** LexVLA outperforms STAIR in most metrics and is comparable in two others, despite using significantly less training data (12M vs. 1.1B). This highlights LexVLA's training efficiency. **4)** The results of $\mathbf{Q}$(BoW) are poor, due to the BoW method limits the model's ability to use information effectively. It struggles with vocabulary [7] and semantic [32] mismatches. LexVLA aims to address these limitations by leveraging language models for lexical prediction rather than relying solely on captioning. **5)** The significant improvements of LexVLA over $\mathbf{Q}$(CLIP) indicate that using DINOv2 as the visual backbone is superior. **6)** LexVLA with FLOPs loss performs worse than our proposed overuse penalty, and activates meaningless tokens as in Fig. 6. In summary, these results demonstrate the effectiveness and efficiency of our method.

## 4.2 Lexical representation analysis

In this part, we evaluate if LexVLA learns the accurate lexical representation. We conduct the following experiments: 1) Quantitatively, we use the proposed PatchDis metric to evaluate the fine-grained patch-level visual lexical representations on MSCOCO 2017 [23]; 2) Qualitatively, we visualize the global lexical representation of images, local lexical representation of image patches, and a PatchDis visualization for the activated patches for the given category.

Table 2: PatchDis results.

| Model | mIoU |
|---|---|
| Random Dis. | 5.0 |
| CLIP | 5.3 |
| VDR | 12.6 |
| $\mathbf{Q}$(CLIP) | 13.9 |
| LexVLA | **36.3** |

**PatchDis evaluation** We report the zero-shot PatchDis mIoU performance on MSCOCO 2017 validation dataset. We compare LexVLA with public available baselines CLIP (0.4B) and VDR (15M), as well as random discrimination, and report the performance in Tab. 2. LexVLA clearly shows the superiority on the patch-level interpretability with the weak global training signal. In contrast, the latent alignment CLIP, due to no supervision on patch-level features, performs slightly better than random guessing. VDR enjoys better interpretability, but significantly worse than our LexVLA. LexVLA variant, using the original CLIP visual model, provides a better and more interpretable

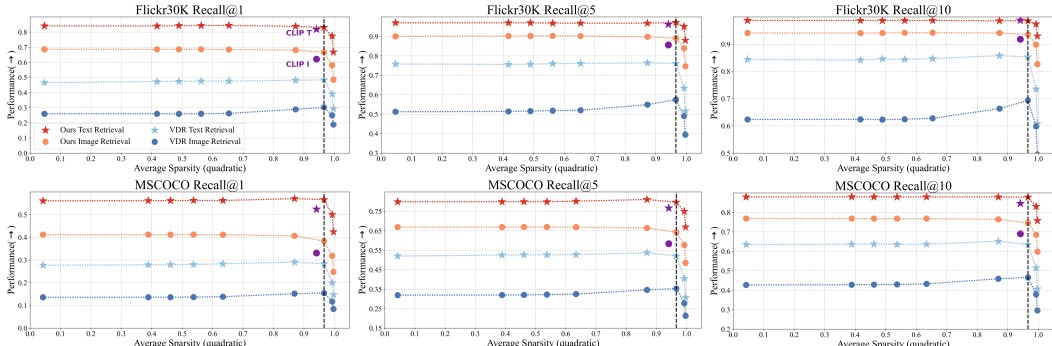

Figure 5: Retrieval in different sparse levels. We compare LexVLA with VDR in the same sparse levels and with CLIP as a proxy of dense latent alignment. The first row is results in the Flickr30K dataset, and the second in the MSCOCO dataset. The first to the third columns show the settings of Recall@1, Recall@5, and Recall@10, respectively. Purple symbols represent CLIP.

lexical representation of visual patches compared to both the original CLIP and previous lexical methods. This further confirms the effectiveness of our proposed approach. We also visualize the predicted patches of different classes in Fig. 3. LexVLA successfully activates the correlated patches in the image for different classes, even for the small-scale objects.

**Lexical visualization**    We visualize the lexical representations of the global image and local patches in Fig. 4. For the bottom row, we randomly select one object in the image and then annotate the patches that cover the object to calculate the lexical representation on patches in the free form. We use word cloud [28] to illustrate the lexical representation where larger word indicates larger lexical value. These results clearly suggest that the learned lexical representation correctly correlates with the semantics of the image/patch, showcasing the effectiveness of LexVLA.

### 4.3   Further analysis

**LexVLA under different sparsity**    One of the targets for lexical representation is to learn a sparse representation for potential benefits in large scale retrieval tasks. To test the robustness of learned representations in LexVLA, we test the retrieval performance of LexVLA in different sparsity levels. Particularly, the sparsity ratio is calculated via $\ell_0(s_i)/V$, the ratio of non-zero elements in the lexical representation $s_i$. We compare LexVLA with VDR on the same sparsity ratios. We also visualize the CLIP's performance as a proxy of dense latent representation.

Results are shown in Fig. 5. **1)** LexVLA is robust against the sparsity ratio even in a very high ratio (98.27%, 296 activated tokens, which is marked with black vertical dotted line in Fig. 5), and then starts to get damaged as the sparse ratio approaches 1; **2)** Compared with VDR, LexVLA achieves better performance in all sparse level, indicating a consistent improvement. **3)** LexVLA enjoys superior performance with less activated feature dimension as CLIP (296 vs 512) in almost all cases, showcasing our model's effectiveness.

**Overuse penalty**    The widely used FLOPs loss aims to encourage the sparsity of learned lexical representation. However, we empirically find that it will falsely activate meaningless tokens as in Fig. 6. Specifically, the model tends to activate certain semantically unrelated tokens across different $s_i$ (like '@' in the image and 'contemporary' in the text, respectively), implicitly treating these tokens as latent dimensions. Our proposed overuse penalty inhibits these frequently activated words more aggressively, and leads to more semantic-correlated lexical representations in Fig. 6. This enhanced interpretability further improves the retrieval performance in most cases, as in Tab. 1.

**Limitations**    The major limitation of LexVLA is the use of vocabulary from large language model. We design our lexical vocabulary based on the Llama 2's tokenizer, and sometimes it splits a word into several sub-word tokens, making the resulting token not a complete word. Although we filter out many meaningless tokens, it still introduces the gap between our vocabulary and the perfect word-level vocabulary. Given that random initialize a word-level vocabulary and additionally learn a

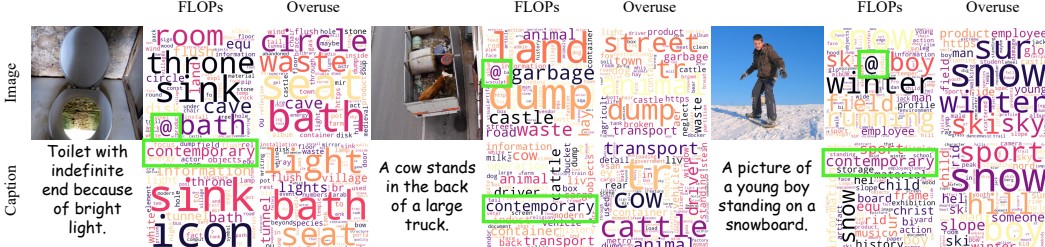

Figure 6: Visualization of lexical representations of LexVLA w/ FLOPs loss and w/ our proposed overuse penalty, respectively. The first row represents images while the second is caption. LexVLA w/ FLOPs loss usually falsely activates similar meaningless tokens frequently as a shortcut (see '@' and 'contemporary' outlined by green boxes as examples). LexVLA w/ overuse penalty significantly mitigate this false activation issue.

projector from sub-word tokens to word-level tokens works poorly, we regard designing a word-level vocabulary that still benefit from the LLMs as a future work.

**Broader impacts**   The advancements made in cross-modal retrieval, as demonstrated in this work, hold significant potential for various fields and applications. One immediate impact lies in enhancing information retrieval systems, where users can efficiently search and retrieve multimodal content such as images and text. This can greatly benefit industries such as e-commerce, digital libraries, and multimedia databases, enabling more intuitive and accurate searches. While the advancements in cross-modal retrieval offer numerous benefits, one concern is related to privacy and security, as the integration of multimodal data may raise issues regarding data protection and user consent. It's essential to mitigate these potential negative impacts through robust privacy measures.

## 5   Conclusion

In this paper, we propose LexVLA, a unified lexical vision-language alignment framework. LexVLA effectively learns a unified lexical representation with unique codebooks for vision and language modalities, with the aid of single-modal pre-trained models. We propose to incremental fine-tune LexVLA with the standard contrastive objective penalized by the proposed overuse objective to prevent meaningless activation and encourage sparse embedding. LexVLA refrains from the complex training configurations, and effectively achieves the patch-level interpretability with only global supervision signals. We introduce a patch-level interpretability metric, PatchDis, to quantify this on the zero-shot cross-modal benchmark dataset. We demonstrate the effectiveness of LexVLA on two zero-shot cross-modal retrieval benchmark datasets with significantly smaller multi-modal training dataset. We also provide detailed analysis on LexVLA.

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

# A Appendix / supplemental material

## A.1 Model hyperparameters

Our hyperparamters selection for the detail can be found in Tab. 3

Table 3: Hyperparameters for training LexVLA.

| Hyperparameter | Value |
|---|---|
| Batch size | 6144 |
| Learning rate | 5e-4 |
| Vocabulary size | 17149 |
| Training epochs | 12 |
| Max temperature $\tau$ | 100.0 |
| Warm-up iterations | 1000 |
| LR Scheduler | Cosine |
| Adam $\beta_1$ | 0.9 |
| Adam $\beta_2$ | 0.999 |
| Adam $\epsilon$ | 1e-6 |
| Penalty $lambda_1$ | 5e-4 |
| Penalty $\lambda_2$ | 1e-3 |
| Lora_alpha | 16 |
| Lora_r | 8 |
| Lora_dropout | 0.05 |

## A.2 Pretraining details of backbone models

The DINOv2 [30] base model used in LexVLA is pretrained with self-supervised learning, leveraging a large dataset of 142 million diverse, high-resolution images. It employs a teacher-student architecture with multi-crop views and strong data augmentations to encourage robustness and capture both global and local features. During pretraining, the model uses a cosine similarity loss between student and teacher outputs to ensure consistent feature representations across different augmentations. This model supports downstream performance on detection and segmentation tasks.

The Llama 2 model [40] used in LexVLA is pretrained with self-supervised learning on approximately 2 trillion tokens from a diverse mix of publicly available datasets and web-sourced content. It employs a dense transformer architecture, and during training, it focuses on token prediction accuracy and generalization. This model supports effective performance across various language tasks.

For more pretraining details, please refer to the original papers of these two pretrained models.

