# OpenReview forum: "Unified Lexical Representation for Interpretable Visual-Language Alignment"
_NeurIPS.cc/2024/Conference — NeurIPS 2024 poster_

### Official Review · Reviewer_CyPa · 2024-07-11

**Soundness:** 3
**Presentation:** 3
**Contribution:** 2
**Rating:** 5
**Confidence:** 4

**Summary:**

The authors propose a method based on lexical representation for Visual-Language Alignment (VLA). The method relies on aligning two strong unimodal models, namely DINOv2 for the visual modality and Llama 2 for the text modality. Each backbone is fine-tuned with a few adapters or additional layers. The two modalities use separate codebooks mapping to a joint vocabulary. The authors also propose an overuse penalty to limit the excessive activation of irrelevant tokens. Finally, the authors introduce the PatchDis metric to measure patch-level alignment. Evaluation on zero-shot cross-modal retrieval datasets shows state-of-the-art performance of the method with the compared baselines. Additional experiments on the patch-level representation and sparsity showing the effectiveness of the method are also reported.

**Strengths:**

- The authors proposed an effective and interpretable Lexical Representation approach for Visual-Language Alignment
- The proposed method is described clearly
- The experimental results show state-of-the-art performance in comparison to the baseline selected

**Weaknesses:**

- The vocabulary is based on the Llama tokenizer which, as stated in the limitations, may split words into meaningless sub-word tokens and may also lack longer relevant words.
- The latent baselines for zero-shot cross-modal retrieval do not include recent methods such as BEiT-3 [Wang, Wenhui, Hangbo Bao, Li Dong, Johan Bjorck, Zhiliang Peng, Qiang Liu, Kriti Aggarwal et al. "Image as a foreign language: Beit pretraining for vision and vision-language tasks." In Proceedings of the IEEE/CVF Conference on Computer Vision and Pattern Recognition, pp. 19175-19186. 2023.]
- One main difference with the compared methods could be the use of the DINOv2 visual backbone and the Llama 2 textual backbone, it is possible the proposed method benefits from these strong backbones. All methods' visual and text backbones (and their potential pretraining) should be discussed in detail to enable the readers to properly judge the merit of the proposed method

**Questions:**

- Have the authors explored a simpler approach of just selecting the nouns, adjectives, and non-auxiliary verbs in a caption instead of the LLM-based lexical predictor? How many keywords are extracted by the LLM on average per caption? Does it vary with the length of the caption?
- Eq (3), what is x with any index?
- Table 1: it would be good to also indicate the amount of (unimodal) pretraining data (if any) used for each method e.g. the amount of data used for DINOv2 and LLama 2 for the proposed method. What are the test splits used for this experiment? Commonly, results are reported based on the splits in [Karpathy, Andrej, and Li Fei-Fei. "Deep visual-semantic alignments for generating image descriptions." In Proceedings of the IEEE conference on computer vision and pattern recognition, pp. 3128-3137. 2015.]. If these are not used it would be good to use them as well.
- Figure 3: it would be good to provide the class <-> color mapping here.
- Figure 4: bottom row, how were the local patches selected?
- Figure 5: what does the vertical black dotted line represent? How were the different sparsity level selected?
- The authors mention in the limitations that their “lexical vocabulary based on the Llama 2’s tokenizer (...) splits a word into several sub-word tokens.” does that also mean that some rather rare long words would not appear in the vocabulary? Have the authors studied what are these missing words? Further down the authors state “Given that random initialize a word-level vocabulary and additionally learn a projector from sub-word tokens to word-level tokens works poorly, we regard designing a word-level vocabulary that still benefit from the LLMs as a future work.”, it seems the author did conduct some experiments towards that. Even if the results were not conclusive it would be interesting to share what was tried and what was the performance.

Typos etc:
- p2-l52: missing space “​​LexVLAto”

**Limitations:**

Yes

---

> ### Author Rebuttal · Authors · 2024-08-07
>
> ## W1. Tokenizer-based vocabulary is not perfect.
> Thanks. Please note that LexVLA has achieved SOTA performance in most experiments, demonstrating its effectiveness. While it is nontrivial to design a perfect vocabulary to handle all the corner cases generally, we would take it as an important future work and already thoroughly discussed in the limitation parts (L289-295). Nevertheless, our main contributions are still well supported by our empirical evidence here, and our work is self-contained.
>
> ## W2. Compare with BEiT-3?
> Thanks for suggesting the paper of BEiT-3 [1]. We will address how it significantly differs from our work (and thus not included in our original submission), while we will cite and discuss it in the related work in the revision. Particularly,
>
> 1) Differences in Goals and Training Strategies:  BEiT-3 uses a masked data modeling strategy to develop a general-purpose model utilizing both single- and multi-modal data; in contrast our approach focuses on developing a specific lexical representation for images and texts.
> As our target is to learn interpretable lexical representation for images/texts, and BEiT-3 has no such interpretable representations, comparing our method with BEiT-3 would not be informative for analyzing the effectiveness of lexical methods.
>
> 2) The comparison between lexical methods and dense methods is to demonstrate that introducing lexical representation does not degrade alignment performance, as shown in the retrieval task. Hence we compare with the contrastive-trained methods, i.e., CLIP and its variants, which are more relevant for assessing the effectiveness of lexical representation.
>
> ## W3. The proposed method possibly benefits from strong backbones?
> Thanks for this comment. But actually, we didn't propose a general alignment method for any two visual and textual backbones. LexVLA does employ the unique properties of DINOv2 and Llama2. Specifically, as highlighted in our first contribution (L68-L70), we utilize single-modal pre-trained models for vision-language alignment tasks to exploit their unique properties, which cannot be fully captured by contrastive objectives alone. The analysis of the results in Sect 4.1 and 4.2 reflects the advantages of the chosen backbone. As we use standard DINOv2 and Llama2, we will elaborate their pretraining details into Supplementary of our paper as suggested.
>
> ## Q1-1. Simpler baseline?
> Thanks for your suggestion. We actually do have explored the approach you suggested, which is one typical and simple method called Bag-of-Words (BOW). The results were poor. This aligns with findings in previous studies [2, 3, 4], hence we did not include these results in our submission. We report results in the rebuttal PDF.
>
> The BOW method limits the model's ability to use information effectively. It struggles with vocabulary [5] and semantic [6] mismatches. LexVLA aims to address these limitations by leveraging language models for lexical prediction rather than relying solely on captioning.
>
> ## Q1-2. How many keywords are extracted? Does it vary with the length of the caption?
> We want to clarify that as stated in Sec. 3.2, when we use a LLM as a lexical predictor, the caption is mapped into a token embedding $y_{text}$, which is then mapped to a lexical representation $s_{text}$.
> The token embedding is not made up of specific keywords but a compact representation.
>
> ## Q3. Details on pretraining data and test splits?
>
> 1) All pretrained models are public released models. Llama 2 is trained on 2 trillion tokens and DINOv2 is trained on 142M images.
>
> 2) Following previous approaches, the test splits are from [7].
>
> We will include these information in the revision.
>
> ## Q4. Questions about Figures.
>
> 1) Fig 3: Thanks. We have added the label mapping in the rebuttal PDF.
>
> 2) Fig 4: In the bottom row, we randomly select one object in the image and then annotate the patches that cover the object to calculate the lexical representation on patches in the free form.
>
> 3) Fig 5: a) Vertical black dotted line: The vertical black dotted line represents the sparse level of our final LexVLA model, which is reported (98.27\%, 296 activated tokens) on P9, L278.
> b) Different sparsity levels: As on P4, L122-L127, sparse levels can be selected via Top-k or Value thresholding. The sparse levels in Fig. 5 were selected using Value thresholding.
>
> ## Q5. Question about lexical vocabulary.
> Thanks, we tokenize all the words in the test set, and found that even the most frequently split words did not appear often, and most of them retained semantics, such as:
> bathroom -> bath, room;
> motorcycle -> motor, cycle;
> sidewalk -> side, walk.
> While corner cases exist, LexVLA has demonstrated effectiveness and achieved SOTA performance in most experiments. We would take it as an important future work and have thoroughly discussed it in the limitations section of our paper (L289-295).
> Please note that our main contributions are well supported, and our work is self-contained.
>
> ## Q2 & Typos
> Thanks!
> The $x$ in Eq(3) should have index $img$.
> We will revise all the typos pointed out by the reviewers in the final version and continue to proofread the manuscript to reduce any potential confusion.
>
>
>
> [1] Wang et al. “Image as a Foreign Language: BEIT Pretraining for Vision and Vision-Language Tasks.” CVPR 2023
>
> [2] Gao et al. "COIL: Revisit exact lexical match in information retrieval with contextualized inverted list." arXiv:2104.07186.
>
> [3] Xiong, et al. "End-to-end neural ad-hoc ranking with kernel pooling." SIGIR 2017
>
> [4] Formal et al. "SPLADE: Sparse lexical and expansion model for first stage ranking." SIC 2021
>
> [5] Croft et al. Search engines: Information retrieval in practice. Vol. 520. Reading: Addison-Wesley, 2010.
>
> [6] Peters, E. et al. “Deep Contextualized Word Representations.” ArXiv abs/1802.05365
>
> [7] Karpathy et al "Deep visual-semantic alignments for generating image descriptions." CVPR 2015

---

> > ### Comment · Reviewer_CyPa · 2024-08-13
> >
> > I have read all the reviews and the responses from the authors, they have addressed some of my concerns and I believe integrating most of the discussion would improve the paper enough to update my rating to `5: Borderline accept`.

---

### Official Review · Reviewer_a8Qd · 2024-07-12

**Soundness:** 3
**Presentation:** 3
**Contribution:** 2
**Rating:** 5
**Confidence:** 4

**Summary:**

The paper proposes LexVLA, a more interpretable VLA framework that learns a unified lexical representation for both modalities without complex design.
LexVLA uses DINOv2 as the visual model and Llama 2 as the language model, proposing an overuse penalty to avoid false discoveries.
LexVLA outperforms baselines on cross-modal retrieval benchmarks, even when fine-tuned on a modest dataset.
Extensive experiments were conducted to analyze LexVLA's performance.

**Strengths:**

1. The paper is easy to follow.
2. The framework does not require complex design or training configurations, making it more accessible and efficient.
3. LexVLA outperforms baselines on cross-modal retrieval benchmarks, even when compared to models trained on larger datasets.
4. Ablation demonstrates the decision choice and effectiveness of proposed components.

**Weaknesses:**

1. I can't quite get the novelty of this work. The lexical representation mentioned in the paper is somehow a way to select important information and then map it to the code book. However, the codebook strategy was explored [1]. Especially the visual part, where does the concept of Lexical come in? Can the author elaborate more on this?
2. In Table 1, the improvement is pretty limited in the bottom block compared to using CLIP in the last and first blocks. It makes readers question whether the performance was gained by the DINOv2 representation.
3. The alignment was tested on only one task, it will be more interesting to test on other multimodal tasks such as zeroshot classification, or even grounding since it has DINOv2 representation.


[1] Duan, Jiali, et al. "Multi-modal alignment using representation codebook." Proceedings of the IEEE/CVF Conference on Computer Vision and Pattern Recognition. 2022.

**Questions:**

Please address the questions in the weakness.

---

> ### Author Rebuttal · Authors · 2024-08-06
>
> ## Q1-1. Is lexical representation a way to select important information and map it to the codebook?
>
> Thank you for your question. We respectfully disagree with this characterization. Lexical representation is not a codebook strategy in [1]. Learning a unified codebook for multi-modal data is fundamentally different from learning a lexical representation. Here’s how they differ:
> 1. **Purpose and Approach:**  As discussed in Sec. 3, our goal is to learn the interpretable lexical representation for images and image patches. In contrast, the codebook strategy [1] aims to learn unified, but uninterpretable and non-explicit-semantic-meaningful codes for image and text.
> 2. **Alignment Mechanism:** As explained in Sec. 3.1, our alignment is conducted at the vocabulary level, not the codebook level. We focus on aligning indexes, which provides more flexibility when training our lexical encoders compared to training codebook learners. In contrast,  the codebook strategy [1] attempts to align the codebook embeddings.
> 3. **Enhanced Interpretability and Efficiency:** As mentioned in the Sec. 2, the lexical representation is widely studied in the field of text information retrieval. This method has been employed in vision-language models due to its good interpretability and high retrieval efficiency. The dense codebook strategy does not meet these properties.
>
>
> [1] Duan, Jiali, et al. "Multi-modal alignment using representation codebook." Proceedings of the IEEE/CVF Conference on Computer Vision and Pattern Recognition. 2022.
>
> ## Q1-2. Where does the concept of Lexical come in regarding the visual part?
> As we introduced on P4, L134-L137, the lexical representation is the output of the visual encoder.  And on P3, L102-L104, this output is a score vector, whose element indicates the similarity between the image and the corresponding word from the vocabulary.
>
> ## Q2. Limited improvement over CLIP?
> Thank you for raising this concern. We would like to clarify that our improvements over CLIP are significant and demonstrate our advantages over competitors. Our main paper and the rebuttal PDF provide comprehensive experiments and analysis to support this. Specifically, as discussed in the analysis of the experimental results (L238-L248)
> 1. **Comparison of different frameworks:** Comparing LexVLA(CLIP) with dense CLIP, our method shows comparable or even better performance. This indicates that our approach achieves well-aligned lexical representations with significantly less multimodal training data (12M) compared to CLIP (1.1B).
> Additionally, our model ensures semantic correctness of the tokens, which is more challenging than merely aligning latent features, as discussed on P1, L32-L36.
> 2. **Further Enhancement with DINOv2 Features:**  Replacing CLIP with the DINOv2 encoder, which provides local-inclined features, further improves our results. This demonstrates that good interpretability contributes positively to retrieval performance. This also aligns with our motivation.
> 3. **Overall Improvement:** Our full model shows substantial improvement over CLIP in our experimental setup, offering superior retrieval performance and better interpretability, as detailed in Table 1 of the rebuttal PDF file.
>
> ## Q3. More interesting to test the alignment on grounding multimodal tasks.
>
> Thank you for your suggestion. Our work primarily focuses on learning interpretable lexical representation for  images, our experiments are extensive and support our claims and contributions. Importantly, our paper does include multiple multimodal tasks. Specifically:
>
> As stated in Section 3.4 and Section 4.2, we proposed a grounding task called PatchDis. This task is inspired by patch-level segmentation but is designed for models that are not trained on fine-grained alignment tasks such as segmentation or detection. Our experiments, as reported in Table 2, demonstrate that our model performs significantly better on this grounding task. This highlights the effectiveness of our approach.
>
> We agree that exploring additional multimodal downstream tasks would be interesting. We are keen to investigate further applications of our proposed LexVLA in the future. Since our current work is self-contained, this could be a valuable direction for future research.
>
> We will clarify these points further in our revision.

---

> > ### Comment · Reviewer_a8Qd · 2024-08-12
> >
> > Thank you for your rebuttal!
> >
> > Upon examining the comparison between the original CLIP model and its modified version, I observed that there is no definitive winner; the original CLIP showed superior performance on MSCOCO, while the new Lexical version excelled on Flickr30k.
> >
> > This outcome is somewhat anticipated, as the aim is to find an exact match between the tokens. However, this might not hold true in retrieval tasks due to the presence of a lot of redundant information in the background. I believe that an alternative evaluation setting is needed to showcase the full potential of this model.

---

> ### Author Response · Authors · 2024-08-13
>
> Thanks for the opportunity to address your further concerns in response to our rebuttal.
>
> ## Regarding the alternative evaluation:
> Thank you for the suggestion. We report the PatchDis evaluation result of LexVLA (CLIP)  as follows:
> | **Model**      | **mIoU**  |
> |----------------|-----------|
> | Random Dis.    | 5.0       |
> | CLIP           | 5.3       |
> | VDR            | 12.6      |
> | *LexVLA (CLIP)*| *13.9*      |
> | LexVLA         | **36.3**  |
>
> These results clearly show that our LexVLA (CLIP) provides a better and more interpretable lexical representation of visual patches compared to the original CLIP and previous lexical method. This further confirms the effectiveness of our proposed approach.
>
> ## Regarding the results in Table 1
> Thank you for your careful review. We would like to supplement that the results in Table 1 demonstrate the effectiveness of LexVLA, even with significantly less multi-modal training data (12M vs. 1.1B).
> Particularly, while CLIP (1.1B) outperforms LexVLA (CLIP) in MSCOCO image-to-text results, it underperforms in MSCOCO text-to-image and Flickr30k image-to-text and text-to-image experiments. Moreover, our full LexVLA model outperforms CLIP (1.1B) in all settings.

---

> ### Author Response · Authors · 2024-08-14
>
> Dear Reviewer a8Qd,
>
> We would like to know if our previous response addressed your concern. We have reported the result of the alternative evaluation in another comment section, which shows LexVLA (CLIP) provides a better and more interpretable lexical representation of visual patches compared to the original CLIP. If there are any additional comments or suggestions you would like to share, we would be more than happy to discuss them further.
>
> Thank you again for your valuable feedback.
>
>
> Best regards,
>
> The Authors

---

### Official Review · Reviewer_KFJh · 2024-07-13

**Soundness:** 3
**Presentation:** 3
**Contribution:** 3
**Rating:** 6
**Confidence:** 3

**Summary:**

This paper presents LexVLA, a vision language alignment method integrating a pretrained vision model and a pretrained language model. To retain the original capabilities of pretrained single-modal models, it adopts a unified lexical representation with unique codebooks. Moreover, the vision model is tuned with a projector, and the text model is tuned with LoRA. A metric for patch-level alignment is proposed to evaluate interpretability. Experiments are conducted on retrieval benchmarks.

**Strengths:**

- The paper is well-written and easy to follow.
- The content is rich. An architecture, an objective, and a metric are proposed.
- Inserting lightweight components to tune vision and language models to learn lexical representation while refraining from original capability degradation is intuitive.
- The LexVLA can be applied to various architectures.
- Experiments are conducted on multiple benchmarks.

**Weaknesses:**

- Even though a new metric is proposed, the effectiveness of its reflection on interpretability is not verified quantitatively or qualitatively.

**Questions:**

- How accurately or reliably does the proposed PatchDis metric evaluate/reflect the interpretability of patch-level visual lexical representations?

**Limitations:**

While multiple technical contributions have been made in this paper, some of the components lack rigorous verification.

---

> ### Author Rebuttal · Authors · 2024-08-06
>
> ## Q1. How accurately or reliably does the proposed PatchDis metric reflect the interpretability of patch-level visual lexical representation?
> Thank you for your concern. We have discussed and analyzed this in the main paper. Our proposed PatchDis is a direct metric for assessing the interpretability of patch-level visual lexical representation. Here’s how:
>
> **1) Interpretability of Lexical Representation:** Lexical representation is inherently interpretable, as each dimension corresponds to the activated value of a word/token in the vocabulary. This has been elaborated on the Introduction (P1-L29) and Related Work (P3-L91), particularly referencing works on lexical representation [1, 2, 3].
>
> **2) PatchDis reflects patch-level interpretability:**
> For a lexical representation of selected patches to be considered interpretable, it should accurately reflect local semantic information without being influenced by non-selected patches or incorrect semantics. In this context, interpretability indicates how well the image patches are represented with clear and understandable semantics.
>
> Our new metric, PatchDis, is explicitly designed to measure how effectively visual patches represent their intended semantics. As discussed in Sec. 3.4 and Sec. 4.2., PatchDis allows us to measure and compare the interpretability of patch-level features across different models, providing a quantitative evaluation of visual feature interpretability at the patch level.
>
> **3) Visualization and Evaluation:** Our visualization of PatchDis, based on the popular MSCOCO 2017 dataset, provides a qualitative reflection of its effectiveness in terms of interpretability.  The segmentation results offer detailed insights into how the model makes matching decisions. This aligns with the definition of interpretability by Biran and Cotton [4], which refers to the degree to which an observer can understand the cause of a decision.
>
> In summary, the evaluation of lexical representation, both quantitative and qualitative, directly reflects this interpretability.
>
> We hope these points address your concern, and we will update these information accordingly if accepted.
>
>
> [1] Formal, Thibault, Benjamin Piwowarski, and Stéphane Clinchant. "SPLADE: Sparse lexical and expansion model for first stage ranking." Proceedings of the 44th International ACM SIGIR Conference on Research and Development in Information Retrieval. 2021.
>
> [2] Luo, Ziyang, et al. "Lexlip: Lexicon-bottlenecked language-image pre-training for large-scale image-text sparse retrieval." Proceedings of the IEEE/CVF international conference on computer vision. 2023.
>
> [3] Zhou, Jiawei, et al. "Retrieval-based Disentangled Representation Learning with Natural Language Supervision." The Twelfth International Conference on Learning Representations. 2024.
>
> [4]  Biran, Or, and Courtenay Cotton. "Explanation and justification in machine learning: A survey." IJCAI-17 workshop on explainable AI (XAI). Vol. 8. No. 1. 2017.

---

> > ### Comment · Reviewer_KFJh · 2024-08-12
> >
> > The authors have addressed my concerns. I will keep my previous rating.

---

### Author Rebuttal · Authors · 2024-08-07

We would like to express our sincere gratitude to all the reviewers for their insightful and constructive feedback on our paper. We are delighted that our work has been positively received, and we appreciate the time and effort each reviewer has put into evaluating our submission.

We are particularly grateful for the recognition of several strengths: our paper’s clarity and ease of understanding (Reviewer KFJh, Reviewer a8Qd, Reviewer CyPa), the rich content and novel contributions including the architecture, objective, and metric (Reviewer KFJh, Reviewer CyPa), and the intuitive approach of inserting lightweight components without degrading original capabilities (Reviewer a8Qd, Reviewer CyPa). The flexibility of the LexVLA framework (Reviewer KFJh, Reviewer a8Qd) and the comprehensive experiments conducted on multiple benchmarks (Reviewer KFJh, Reviewer a8Qd) were also appreciated. Additionally, the straightforward design and efficiency of our framework (Reviewer a8Qd), its superior performance on cross-modal retrieval benchmarks (Reviewer KFJh, Reviewer CyPa), and the clear demonstration of the effectiveness of our proposed components (Reviewer KFJh, Reviewer a8Qd) were well-received. Lastly, we are pleased that our method’s effectiveness and interpretability and the state-of-the-art performance compared to baselines (Reviewer CyPa) have been acknowledged.

We also extend our gratitude to the Area Chair for their guidance and support throughout the review process.

Thank you once again for your valuable feedback.

---

> ### Comment · Area_Chair_apT4 · 2024-08-10
>
> Dear reviewers:
>
> There are diverging reviews for this paper. Supporting reviewers, please take a look at the other reviews. For the opposing reviewer, CyPa, please carefully read the rebuttal and share any remaining concerns, if any.
>
> Thanks

---

### Decision · Program_Chairs · 2024-09-25

**Decision:**

Accept (poster)

**Comment:**

After discussion, this submission received 3 positive scores. After rebuttal, the major concerns about interpretability of the proposed approach, result visualization, experimental evaluation, and comparison of difference frameworks were comprehensively solved. After reading the paper, the review comments and the rebuttal, the AC think the remaining concern is about the lack of critical evaluation of the proposed approach, which is encouraged to added to the camera-ready version.